# Effectiveness of the Pelvic Clock and Static Bicycle Exercises on Wisconsin Gait Scale and Trunk Impairment Scale in Chronic Ambulatory Hemiplegic Patients: A Single Group Pre-Post Design

**DOI:** 10.3390/healthcare11020279

**Published:** 2023-01-16

**Authors:** Thamer Altaim, Venkatesan Ramakrishnan, Shenbaga Sundaram Subramanian, Sugasri Sureshkumar, Makesh Babu Subramanian, Vijayamurugan Eswaramoorthi, Riziq Allah Gaowgzeh, Saad Alfawaz, Ziyad Neamatallah, Umar Alabasi, Majed Albadi

**Affiliations:** 1Physical Therapy Department, Faculty of Allied Medical Sciences, Aqaba University of Technology, Aqaba 11191, Jordan; 2Chettinad School of Physiotherapy, Chettinad Hospital and Research Institute, (CHRI), Chettinad Academy of Research and Education (CARE), Kelambakkam 603103, India; 3Faculty of Physiotherapy, Meenakshi Academy of Higher Education & Research, Chennai 600078, India; 4Sree Anjaneya College of Paramedical Sciences, Malabar Medical College Hospital & Research Centre Campus, Calicut 673315, India; 5Faculty of Health Science, School of Rehabilitation Science, Universiti Sultan Zainal Abidin (UniSZA), Gong Badak Campus, Kuala Nerus 21300, Malaysia; 6Department of Physical Therapy, Faculty of Medical Rehabilitation Science, King Abdulaziz University, Jeddah 21589, Saudi Arabia

**Keywords:** pelvic clock exercise, static bicycle, chronic hemiplegic, gait performance, trunk performance

## Abstract

Background: Most Hemiplegic patients achieve ambulatory function during the sub-acute stage of stroke. Though ambulatory, they still perform an unpleasant awkward gait with remarkable compensations requiring more energy expenditure. Fatigue arises at an early duration as a result of increased energy expenditure. The walking pattern becomes circumduction, featured by asymmetry with an extensor synergy of the lower limb. Each step is rotated away from the body then towards the body, forming a semicircle. This leads to changes in various parameters of gait (spatiotemporal, kinematic, and kinetic) in hemiparetic patients. Purpose: Many studies reveal the effectiveness of various therapeutic techniques in managing hemiplegic circumduction gait. Pelvic clock exercises aid in improving pelvic rotation components and cause dissociation in impaired pelvic mobility due to spasticity. A static bicycle helps in enhancing proper control between the hamstrings and quadriceps. It also helps in improving knee flexion range. As the patient places the foot in the cycle’s petals, it helps to enhance dorsiflexion and eversion functions as well. As the lower body is exercised, there could be relative changes in the upper body, i.e., the trunk. Thus, this study aimed to determine the changes in gait functions and trunk performance of chronic ambulatory hemiplegic patients in response to the above therapies for four weeks. Method: Twenty-five subjects (post-stroke duration (2.8 ± 0.6) years) who could walk 10 m independently without assistance or support of aid participated in a pelvic clock and static bicycle exercise intervention. The session duration was 30 min a day, and therapy was delivered six days a week and continued for four weeks. The entire program was carried out in an outpatient neurorehabilitation center. Results: After the intervention with pelvic clock and static bicycle exercises, there was a remarkable change in gait and trunk functions in chronic hemiplegic patients. Conclusion: The exercises comprising pelvic clock and static bicycle showed positive differences in gait and trunk functions in chronic stage hemiplegic patients. Later, randomized controlled studies involving larger sample sizes, advanced activation techniques, and increased intervention duration will explore in-depth information on their effectiveness and clinical significance.

## 1. Introduction

The assessment of proper gait and trunk performance in stroke patients is challenging [1]. Various clinical scales and advanced devices were developed to monitor gait and trunk performances, such as movement quality, pattern, sequence, and muscle tone alteration [2]. Sensory and motor loss due to stroke varies broadly in severity based on brain area involvement [3]. The existence of concurrent disorders such as apraxia complicates clinical features [4]. Gait and trunk assessment and recovery are critical because mobility increases the chances of activity participation and community re-entry [5]. The activity of walking is described as joint kinematics and muscular forces. There are permissible degrees of variation in the gait of normal individuals [6]. Hemiparetic patients’ gait characteristics differ with involved brain areas such as the motor cortex, basal ganglia, and cerebellum and the variety of trick movements the patient develops during their sub-acute stage [7]. One thorough literature review represented numerous studies on hemiplegic gait, concentrating on identifying logical and practical methods to evaluate and notice changes in gait. Some research works assessed the utility of the Wisconsin gait scale (WGS) in various rehabilitation phases. WGS was an exclusive scale developed to examine the gait of hemiplegic patients [8]. The tool consisted of 14 items measuring kinematics and temporal and distance gait parameters, which are the standard parameters that are deviated in hemiplegic patients. The key gait determinants are the rotation of the pelvis, the tilting of the pelvis, the lateral displacement of the pelvis, the flexion of the knee, the flexion of the hip, and the dorsiflexion of the ankle. Working back on these six determinants of gait in hemiplegic patients has an impact on the quality of gait.

Similar to gait, trunk activity helps to better stabilize and balance the right and left sides. Trunk activity helps to minimize the trunk spasticity associated with the deeper muscles. A flexible trunk has a better balance reaction compared to a spastic trunk [9]. Pelvic clock exercises, when introduced in hemiplegic patients, help improve pelvic rotation and tilt pattern. Improved pelvic rotation and tilt have a relative reduction of tone in the trunk [10].

Similarly, exercises involving a static bicycle enhance hip and knee flexion and improve the muscular balance between the quadriceps and hamstrings [11]. The placement of the foot in the stationary bicycle pedals improves joint proprioceptors and performs corrections in ankle dorsiflexion and eversion [12]. With the activity of lower limbs, the stabilization component of the trunk will become activated. The trunk impairment scale (TIS) evaluates static and dynamic balance and trunk coordination in sitting. Numerous studies used the TIS as an outcome scale for noticing changes in trunk performance. The TIS is reliable, valid, and internally consistent for use in clinics and research. Most hemiplegics in the chronic stage have come across various rehabilitative components in their acute and sub-acute settings. Giving due consideration to their interest and self-motivation, a 4-week intervention duration was adopted in this study. They identified a statistically reliable change in walking and trunk performance. Therefore, the main aim of this pre-post design study was to determine the effectiveness of 4 weeks of pelvic clock and static bicycle exercises in the trunk and walking activities in chronic ambulatory hemiplegic patients utilizing theTIS and WGS, respectively.

## 2. Materials and Methods

We studied 25 hemiparetic patients of the age range between 45 and 62 years (mean age 53.48 years). Inclusion criteria were age > 45 years. The walk-ins to the outpatient center were of the above age groups, were one-side hemiplegics involving one lower limb, had the cognitive ability to follow instructions for performing the gait analysis and exercises (MoCA Score above 26), and had the ability to walk 10 m without assistance (Brunnstrom’s recovery stages 3 to 5). Exclusion criteria were patients with bilateral stroke or recurrent stroke or with a history of existing neurological, musculoskeletal, or cardiorespiratory disorders other than stroke. Thirteen patients were right-sided hemiplegics, and 12 were left-sided hemiplegics. The duration from stroke onset to admission to rehabilitation was between 0.8 and 2.3 years (mean 16.23 months). Twelve patients had lesions in the cortex, and 13 had them in the basal ganglia. The cause for the cerebrovascular event was thromboembolism in 21 and hemorrhage in 4. All the patients had one stroke attack and could walk independently without assistance for a minimum of 20 m. All participants received physiotherapy at the department of physiotherapy, Prime Ortho and Shoulder Care, on an outpatient basis. The rehabilitation program was tailored to the individual, having a normalization of movements and minimization/elimination of abnormal compensatory movements as the goal. Treatment was tuned, keeping Body Mass Index (BMI), aerobic capacity, individual postural control status, and upper and lower limb functions in consideration. The intervention consisted of four components.

First is the warm-up component. Patients were trained on mat activities such as rolling and sitting up and then attempting to stand without assistance. The training emphasized obtaining normal passive range in the lower limbs, balance activities, weight shifting, and improving pelvic rotation and forward initiation.

The second component was the pelvic clock exercise (Figure 1). The pelvic clock exercise consisted of 10 repetitions of lateral pelvic tilts followed by ten repetitions of anterior and posterior pelvic tilts. The patients in a crooked lying position performed all the tilts. A pelvic clock device was placed below the sacral region to facilitate the desired tilts. The exercise progressed with appropriate trunk PNF techniques such as resisted pelvic anterior elevation with posterior depression.

The third component was the static bicycle exercises (Figure 2). The bicycle seat height was altered according to the individual patient’s height in centimeters. Patients were asked to sit over the seat comfortably. They were asked to perform slow pedaling in a forward direction for 50 repetitions and backward direction for 50 repetitions. Their upper trunk was maintained in a neutral position without compensatory leaning to the maximal possible extent.

In addition to the above intervention, the fourth component, standard conventional therapy, was given for the trunk and upper extremities. The total intervention duration was 45 min (5 min warm-up, 15 min of pelvic clock and static bicycling, and 15 min of standard conventional therapy for trunk and upper extremities, with 5 min of rest in between each exercise maintained for ensuring the quality of exercises). Pelvic clock exercises help break the stiffness associated with the pelvic girdle, and static bicycle exercises help improve hip and knee flexor activation. As these exercises have direct address on gait determinants, these became the rationale for choosing them as the main interventional exercises

The outcomes were assessed in every patient pre- and post-intervention.

WGS has 14 variables measuring clinically relevant gait components. The variables test temporal parameters and body movement patterns in all gait phases. Rodriques et al. developed this scale. It is a reliable and valid measure for hemiplegics. As it is a visually observed scale, affected side parameters are scored by comparing them to normal side parameters. The minimum (best) possible WGS score is 14, and the maximum (worse) likely is 45. The participants were evaluated by the same physiotherapy consultant during walking without the help of supportive devices but were permitted to use a walking aid. The flexor synergy was compromised and was a collection of movements such as hip flexion, abduction, external rotation, knee flexion, ankle dorsiflexion, and sub-talar inversion and eversion.

Verheyden proposed the TIS scale in the year 2006, and TIS examines sitting balance and trunk coordination. There are two components, static and dynamic. The total points for TIS can be 0 for a poor functional performance and 23 for an excellent functional performance.

### Statistical Analysis

The Kolmogorov-Smirnov test was used to assess the normality of data distribution. Data were presented as mean and standard deviation (SD). Within-group differences from pre-test to post-test measurements were evaluated using paired *t*-tests. A *p*-value < 0.05 was considered to be statistically significant. The contrast between WGS and TIS scores before and after intervention was assessed using paired *t*-tests. The Karl Pearson correlation coefficient method was used to establish the intensity of the linear relationship between WGS and TIS for pre- and post-intervention. The R-value was calculated, and a *p*-value < 0.05 was considered statistically significant.

## 3. Results

The total participants were 25 patients (Table 1). Eight were aged between 45 and 50, 16 patients were between 51 and 60, and 1 patient was between 61 and 70. Nineteen of them were male, and 6 of them were female (Table 2). Based on artery infarction, 11 patients were with ACA, four patients were with MCA, four patients were with PCA, three were with VBA, and three were with the cerebellar artery (Table 3). Based on the duration of the incidence of stroke, six patients were less than one year, 17 patients were between 1 and 2 years, and two patients were between 2 to 3 years (Table 4).

Figure 3 shows that the average of TIS scores was 9.75 before the commencement of the intervention and the average of TIS scores was 15 after the providing the intervention there was an increase in the trunk impairment scale from a pre-intervention average of 9.752 to a post-intervention average of 15 (Figure 1). The *p*-value 1.914 × 10−12 was <0.00001, which indicates a statistically significant improvement from pre- to post-intervention (Figure 3). There was a reduction in the Wisconsin gait score from a pre-intervention average of 36.298 to post-intervention 32.538 (Figure 4). The *p*-value 1.362 × 10^−10^ was <0.00001, which indicates a statistically significant improvement from pre- to post-intervention.

There was a positive correlation between the pre-intervention scores of WGS and TIS (Figure 5), i.e., the increase in the Wisconsin gait score was noticed with an increase in the trunk impairment scale pre-intervention (Figure 6). The value of R was 0.2407. Although technically a positive correlation, the relationship between variables was weak (the nearer the value was to zero, the weaker the relationship). The *p*-value was 0.246446. The result was not significant at *p* < 0.05. On correlating the post-intervention scores of WGS and TIS, there was a negative correlation, i.e., the decrease in the Wisconsin gait score was noticed with an increase in the trunk impairment scale post-intervention (Figure 7). The value of R was −0.5445.

There was a moderate negative correlation, which means that there is a tendency for high Y variable scores to go with low X variable scores (and vice versa). The value of R2, the coefficient of determination, was 0.2965. The *p*-value was 0.004938. The result was significant at *p* < 0.05. The study interpretation showed that the WGS score and TIS score were worthwhile at capturing walk quality and trunk quality increments in response to pelvic clock and static bicycle exercises. The significance of progress on the total WGS and TIS scores was high. Highly improved variables were the pelvic rotation, hip flexion, and knee flexion parameters in WGS and components 3 of static and 4 of dynamic sitting in TIS. The differed scores witnessed incremental effects after the intervention program; the results suggested that the WGS and TIS were more closely linked between the trunk and lower limb functional recovery.

## 4. Discussion

Research shows that gait parameters are vital in evaluating collective walking execution in hemiplegia. In this study, we interpreted remarkable enhancement in gait performance between pre- and post-intervention. Walking performance post-intervention was correlated with modified Ashworth’s spasticity grading and Brunnstrom’s recovery stage. Rodriques et al., in their research, also witnessed WGS showing performance profits in 18 stroke patients walking before and following intervention [13]. Altered trunk position sense and trunk muscle weakness influence balance in hemiplegics [14,15]. Postural adjustments of trunk muscles anticipated during functional activities are essential in establishing antigravity postures [16].

On sitting balance recovery, side-to-side stability and balance were more involved than front-to-back stability and balance. Lateral stability and balance control strongly related to balance assessed with the Berg balance scale. This explored the association between strong leg muscle strength and trunk stabilization, especially in the forward-to-backward direction, but side-to-side sitting balance relies purely on the trunk [17]. Fall risk results in deficient functional performance due to poor balance [18]. Exercises in the form of circuit training and the state of task-oriented training help in improving functional mobility [19]. These pelvic clock exercises and static bicycling suit circuits and task-oriented forms of exercise to an extent. Pieces of evidence support that the balance and gait abilities of hemiplegics coordinate with the trunk abilities based on TIS [20]. Functional activity predictors such as the trunk control test, sitting test, sitting balance scale, and trunk impairment scale during discharge and follow-up OPD reviews recorded incremental/detrimental effects. The hospital-based rehabilitation duration in hemiplegic patients with poor trunk performance was lengthy compared to that of subjects with better initial trunk function scores. There was a relative difference in their respective walking distance and ability [21]. Independent sitting balance preceded independent ambulation. With this fact, the rate of independence in all activities can be predicted [22]. Trunk rehabilitation is very much indicated in stroke patients. Patients’ awareness of symmetry with proper feedback enhances trunk balance and forms the basis for reach-out activities [23]. The corrective exercise strategy programs enhance stable scapula, and they work on improving posture. Improved trunk and gait performances lower the NIHSS Score. The lower the NIHSS score, the more melancholy, and the higher the NIHSS score, the higher the chances of becoming dependent [24].

Using appropriate trunk PNF techniques such as resisted pelvic anterior elevation with posterior depression benefit walking initiation. Lateral trunk muscle performance is also enhanced with alternating weight transference, which can be practiced during initial sub-acute phases [25]. A current randomized controlled trial on early trunk exercises after stroke noticed enhanced balance and gait functions [26]. Trunk exercises with a Swiss ball produced quality trunk control, especially in rotation and dynamic balance performance in hemiplegic survivors [27]. The provision of trunk rehabilitation in chronic hemiplegics also proved enhanced balance and gait performances [28]. The WGS assesses the hemiplegic gait deviations in detail. It is essential to arrange these deviations following the stance and swing phases of the gait cycle. Similar to WGS, we also found significant improvement in TIS trunk functions. It was well correlated with the ability of trunk rotation and diagonal forward bending towards the affected and unaffected foot. The symmetry of the trunk is also achieved with the help of the mentioned interventions. The results of this four-week duration intervention were suggestive of positive changes; thus, extending the intervention duration to 6 to 8 weeks will yield better results.

## 5. Conclusions

The exercises comprising pelvic clock and static bicycle showed positive differences in gait and trunk functions in chronic stage hemiplegic patients. In the inference of our identifications, there were marked changes in WGS components such as (a) use of handheld gait aid, (b) step length of unaffected side, (c) guardedness, (d) circumduction at the mid swing, and (e) pelvic rotation at terminal swing. We endorse that WGS can be used in addition to other functional assessment scales while assessing hemiplegic gait. The scale provides in-depth details about standards and deviations of gait. It ideally determines the attitude of body parts in all gait phases and thus showcases the diseased patterns that patients adopt during walking.

There were similar changes in the trunk impairment scale, especially in static sitting, dynamic sitting, and coordination. Using the WGS and the trunk impairment scale is a practical way to witness improvements in gait quality over recovery. We also recommend TIS for noticing relative changes in trunk functions in response to the mentioned therapy.

Limitation: The results suggest positive changes from pre- to post-intervention. However, this study had a minimal sample size and limited range of age groups (45–62) and a limited duration from the onset of stroke. Individual participant’s medication differences were not considered. Specific and accurate outcome measures such as EMG, gait lab report parameters compared to WGS, and TIS assessing feasibility can be utilized if feasible.

Recommendations: Later randomized controlled studies involving a larger sample size, advanced activation techniques, and increased intervention duration will explore in-depth information on its effectiveness and clinical significance

## Figures and Tables

**Figure 1 healthcare-11-00279-f001:**
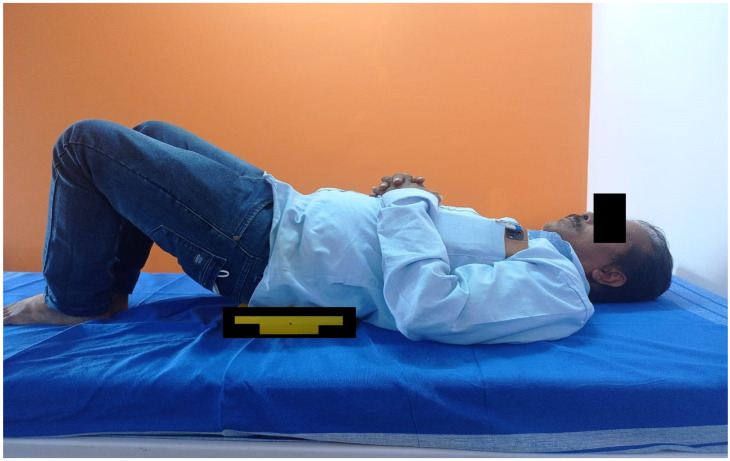
Patient performing pelvic clock exercises.

**Figure 2 healthcare-11-00279-f002:**
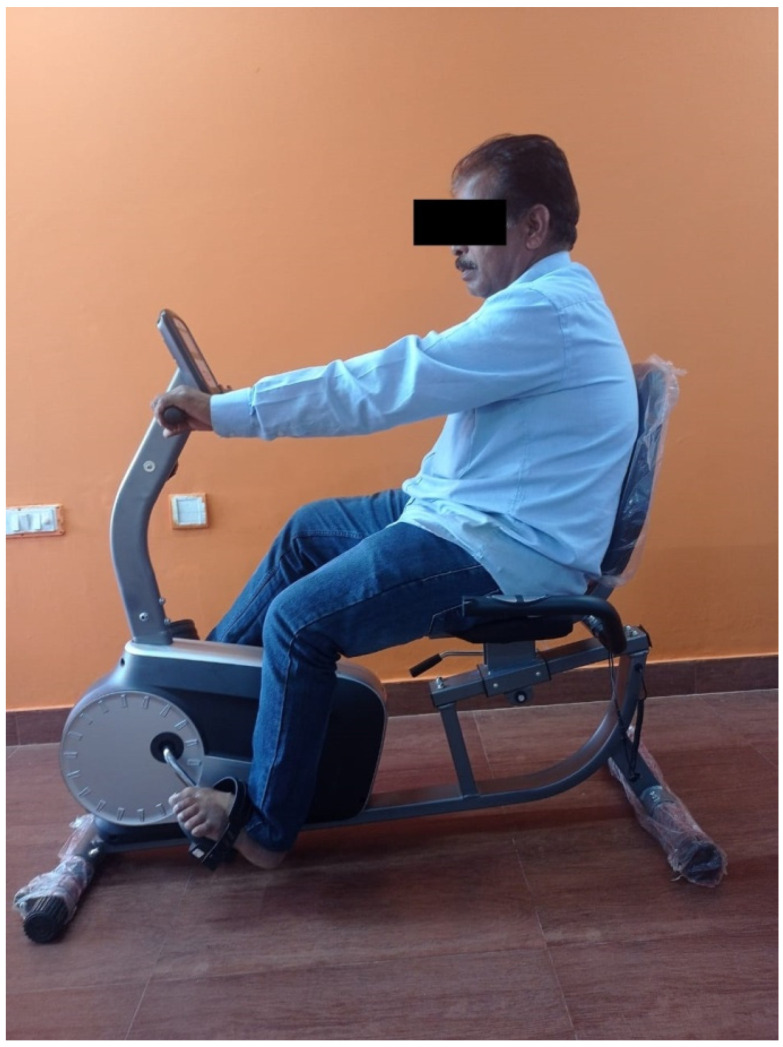
Patient performing static bicycle exercises.

**Figure 3 healthcare-11-00279-f003:**
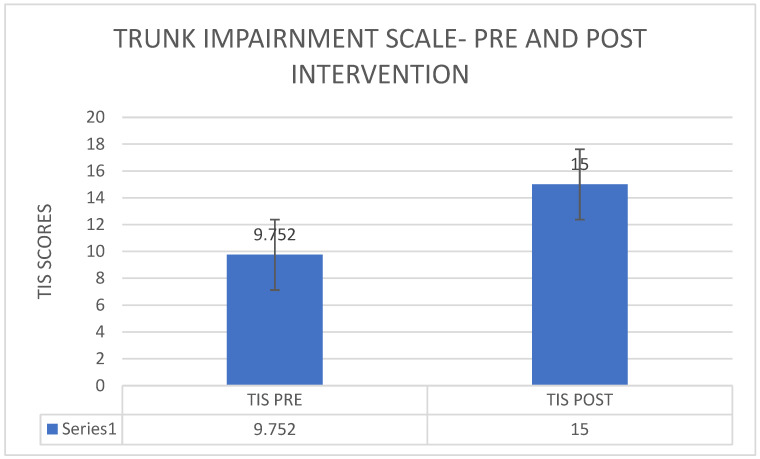
Pre–post comparison of trunk impairment scale.

**Figure 4 healthcare-11-00279-f004:**
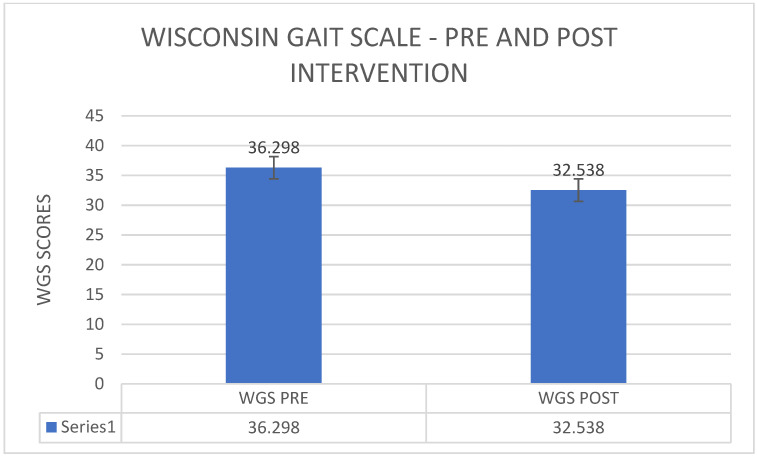
Pre–post comparison of Wisconsin gait scale.

**Figure 5 healthcare-11-00279-f005:**
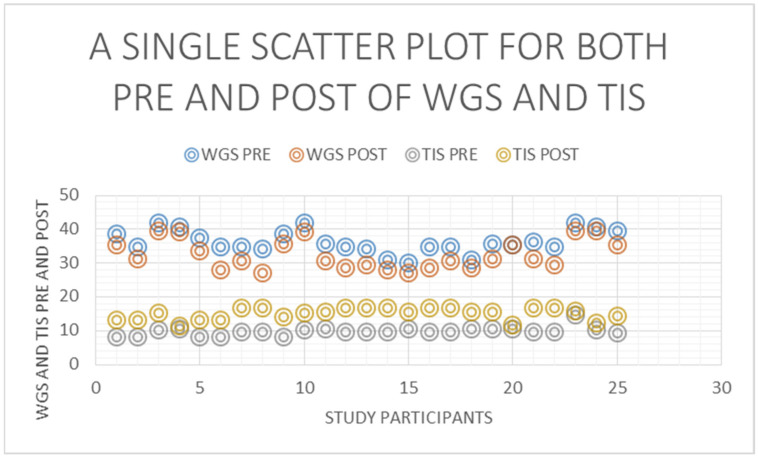
Scatter plot for pre and post data for both trunk impairment scale and Wisconsin gait scale-2 axis (Y-Y).

**Figure 6 healthcare-11-00279-f006:**
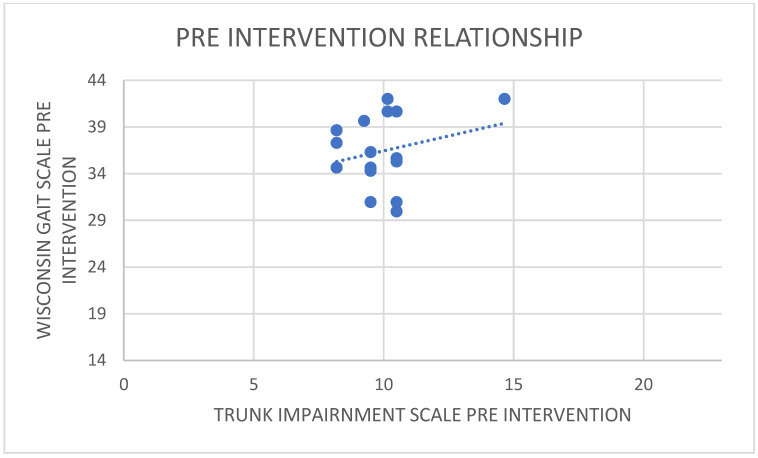
Relationship between pre trunk impairment scale and pre Wisconsin gait scale (weak positive correlation).

**Figure 7 healthcare-11-00279-f007:**
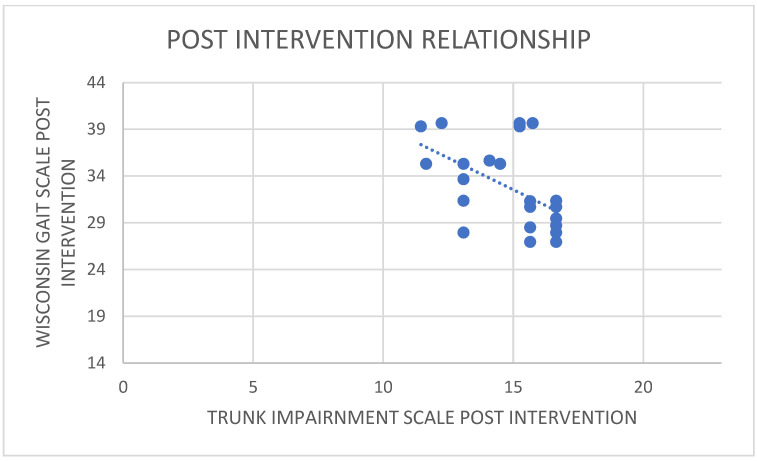
Relationship between post trunk impairment scale and post Wisconsin gait scale (moderate negative correlation).

**Table 1 healthcare-11-00279-t001:** Age distribution of participants.

Age Range	Number of Patients	Percentage
45–50	08	32%
51–60	16	64%
61–62	01	4%

**Table 2 healthcare-11-00279-t002:** Gender distribution of participants.

Gender	Number of Patients	Percentage
Male	19	76%
Female	06	24%

**Table 3 healthcare-11-00279-t003:** Artery involved in participants.

Artery	Number of Patients	Percentage
ACA	11	44%
MCA	04	16%
PCA	04	16%
VBA	03	12%
Cerebellar	03	12%

**Table 4 healthcare-11-00279-t004:** Duration from onset of stroke in participants.

Duration	Number of Patients	Percentage
less than 1 year	06	24%
between 1 and 2 year	17	68%
between 2 and 3 year	02	8%

## Data Availability

The data presented in this study are available on request from the corresponding author. The data are not publicly available due to privacy and ethical restrictions.

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
