# Peer review of "Effectiveness of the Pelvic Clock and Static Bicycle Exercises on Wisconsin Gait Scale and Trunk Impairment Scale in Chronic Ambulatory Hemiplegic Patients: A Single Group Pre-Post Design"

_healthcare, 2023, doi:10.3390/healthcare11020279_

Round 1

Reviewer 1 Report

I studied your manuscript entitled “Effectiveness of the pelvic clock and static bicycle exercises on Wisconsin gait scale and trunk impairment scale in chronic ambulatory hemiplegic Patients: A Single group Pre-Post Design”. Although the approach is interesting, some spaces need to be improved in terms of journal quality. I recommend minor revision before further consideration for publication in the “Healthcare”.

1. What would be the reasons for the therapy to last four weeks? Do the authors predict that extending the duration of therapy would result in a significant improvement in gait and trunk functions in chronic hemiplegic patients?

2. What was the reasoning in choosing inclusion criteria of: age > 45 years?

3. What was the reasoning in choosing pelvic clock and static bicycle exercises?

4. Research design and report (in terms of description) seems to be of good quality. However my greatest critic concerns graphical representation. First, the figures are not referenced in the text. Second, captions are formatted incorrectly, as are numeric values (for example what does "08%" and "01%" on Fig. 3 means exactly, also the graph doesn't really represent his values). So the graphs really need borders? Bar charts do not have labeled axes. Graphs marked as 3 and 4 are completely illegible and need to be redone.

5. Putting additional effort into text formatting is highly recommended. There are a lot of editing errors in the manuscript, such as lack or double space signs (e.g. lines 36, 40, 42, 64 and so on)

6. The text should be reviewed thoroughly to remove some incorrect or confusing phrases. English proofreading is strongly recommended.

The authors need to address the above mentioned points for the betterment of the manuscript.

Author Response

  1. What would be the reasons for the therapy to last four weeks? Do the authors predict that extending the duration of therapy would result in a significant improvement in gait and trunk functions in chronic hemiplegic patients?

Response: Most of the hemiplegics in chronic stage have come across variety of rehabilitative components in their acute and sub-acute stages. Giving due consideration to their interest and self- motivation a 4 week intervention duration was adopted in this study.

The results of this 4 week duration intervention were suggestive of positive changes, so extending the intervention duration to 6 to 8 weeks will definitely yield better results.

       2.What was the reasoning in choosing inclusion criteria of: age > 45 years?

 Response: There is relatively lower number of patients with stroke at younger age .Presence of causative co morbidities is higher at age category above 45 years

      3.What was the reasoning in choosing pelvic clock and static bicycle exercises?

Response: Pelvic clock exercises helps in breaking the stiffness associated with pelvic girdle and static bicycle exercises helps in improved hip and knee flexor activation. As these exercises have direct address on gait determinants, these became the rationale for choosing them as main interventional exercises

 4. my greatest critic concerns graphical representation. First, the figures are not referenced in the text. Second, captions are formatted incorrectly, as are numeric values (for example what does "08%" and "01%" on Fig. 3 means exactly, also the graph doesn't really represent his values). So the graphs really need borders? Bar charts do not have labeled axes. Graphs marked as 3 and 4 are completely illegible and need to be redone.

Response: Figures are referenced to text, captions are formatted,percentage is corrected,borders are added to graph,labelled axes are added & graph 3 and 4 were redone with scatter plots 

    5. Putting additional effort into text formatting is highly recommended. There are a lot of editing errors in the manuscript, such as lack or double space signs (e.g. lines 36, 40, 42, 64 and so on)

Response: Text Formatting was done to make sure spaces between words are correct

6.The text should be reviewed thoroughly to remove some incorrect or confusing phrases. English proofreading is strongly recommended.

Response:Possible corrections were done to remove incorrect and confusing phrases.English proof reading was performed with native english professional

Reviewer 2 Report

The general study design and method are appropriate and the intervention is well-designed and defined.

The paper contains several grammatical mistakes and has a significant amount of repetition that should be addressed. Methods should not contain a discussion of the relative value of the methods, especially since this is already established in the introduction. 

Descriptive parameters of the study population would be more appropriately described in a table, rather than a series of pie charts.

Results should include full descriptive statistics as well as p values for the statistical analysis. I could not find the results of the statistics in the paper. Standard error bars or standard deviation bars should be added to the pre-post overall mean bar charts, I recommend one figure with both pre and post for both conditions, a 2-axis (y-y) bar graph could be easily used. 

Correlation analysis should use x-y scatter plots to show the relationship between outcomes. I would recommend a single scatter plot for both pre and post-data using the same data as given in the draft. This would show correlation and interventional impact well. 

Author Response

1)The paper contains several grammatical mistakes and has a significant amount of repetition that should be addressed. Methods should not contain a discussion of the relative value of the methods, especially since this is already established in the introduction. 

Response:  grammer corrections were made, repetitions were removed, portions of methodology stating established facts were removed

2)Descriptive parameters of the study population would be more appropriately described in a table, rather than a series of pie charts.

Response: pie charts were replaced with tables

3)Results should include full descriptive statistics as well as p values for the statistical analysis. I could not find the results of the statistics in the paper. Standard error bars or standard deviation bars should be added to the pre-post overall mean bar charts, I recommend one figure with both pre and post for both conditions, a 2-axis (y-y) bar graph could be easily used. 

Response: descriptive statistics with p value and R value added in results,statistics added,error bars added to bar charts for pre and post,a 2 axis yy bar is added

4)Correlation analysis should use x-y scatter plots to show the relationship between outcomes. I would recommend a single scatter plot for both pre and post-data using the same data as given in the draft. This would show correlation and interventional impact well. 

Response: x-y scatter plots to show the relationship between outcomes were added.pre pre and post post scatter plots were added with same data

Round 2

Reviewer 2 Report

There doesn't appear to be any attempt to remove redundancy, in-fact I don't see anything taken out, only new content added. Some of which is again redundant with what is already provided. 

The paper still contains a large number of formatting / grammar errors. For example:
+ In the list of authors, and in the body of the paper, many of the entries do not have a space after punctuation.
+ In the introduction 'mostor loss due to stroke [v]ary broadly in [severity?] based on the degree of the involved area.'
+ Seems to be using 'degree' and 'involved' to imply a variance in location.
+ You seem to imply that co-morbidities are too high at ages over 45, but you only recruited between ages 45 and 62. Don't make up inclusions criteria, just tell us what you did. 

Pie charts were not replaced (as stated in the reply to authors), they were made bigger. No table was added.

If you're looking at ages 45-62, why use 41-50, 51-60, and 61-70 as you age bins?

Authors said that they added a y-y plot, but they did not.

Author Response

Respected reviewers,
As suggested by you, the following revisions are applied to the earlier manuscript. The comments and
respective revisions are listed below.

Comments and Suggestions for Authors
1) There doesnt appear to be any attempt to remove redundancy, in-fact I dont see anything taken out,
only new content added. Some of which is again redundant with what is already provided.
Response-Attempts to remove repetition of sentences and facts were done. New content added in response to 1st  round was reframed.
2) The paper still contains a large number of formatting / grammar errors. For example:
+ In the list of authors, and in the body of the paper, many of the entries do not have a space after punctuation.
+ In the introduction mostor loss due to stroke [v]ary broadly in [severity?] based on the degree of the involved area.
+ Seems to be using  to imply a variance in location.
Response-Formatting, especially space after punctuations were done, corrections in Introduction made.
Implication of degree and involved rewritten, Grammar corrections was done
3) You seem to imply that co-morbidities are too high at ages over 45, but you only recruited between
ages 45 and 62. Don't make up inclusions criteria; just tell us what you did.
Response-Walk in patients to our Rehabilitation center were of age group between 45 and 62,so we did the research in accessible patients
4) Pie charts were not replaced (as stated in the reply to authors), they were made bigger. No table was added.
Response-Pie charts were replaced with Tables
5) If you are looking at ages 45-62, why use 41-50, 51-60, and 61-70 as you age bins?
Response-Age bins changed as 45-50, 51-60 and 61-62
6) Authors said that they added a y-y plot, but they did not.
 Response-(Y-Y) scatter plot is also added.
